# Community perception towards mental illness and help-seeking intention in Southwest Ethiopian Peoples Regional State

Dawit Getachew[1]*, Gebremeskel Mesafint[2], Nahom Solomon[1], Kidus Yenealem[3], Zenebu Muche[4], Sewagegn Demelash[4]

1 Department of Public Health, School of Public Health, College of Medicine and Health Sciences, Mizan Tepi University, Mizan Aman, Southwest Ethiopia Regional State, Ethiopia, 2 Department of Psychiatry, College of Medicine and Health Sciences, Mizan Tepi University, Mizan Aman, Southwest Ethiopia Regional State, Ethiopia, 3 Department of Social Work, College of Social Sciences and Humanities, University of Gondar, Gondar, Amhara National Regional State, Ethiopia, 4 Department of Psychology, Institute of Educational and Behavioral Science, Debre Markos University, Debre Markos, Amhara National Regional State, Ethiopia

☯ These authors contributed equally to this work.

* getdawit2011@gmail.com

**Data Availability Statement:** All relevant data are within the manuscript and its Supporting Information files.

## Abstract

### Background

Community perception of mental illness is a collective belief system and attitude about mental disorders; it affects the availability of services, the level of stigma, and the help-seeking intention. This study assessed community perceptions towards mental illness and help-seeking intentions in Southwest Ethiopia.

### Methods and material

A community-based analytical cross-sectional study was done in Southwest Ethiopian People's Regional State (SWEPRS), from March 1st to June 30th, 2021. All adult individuals >18 years old living in the region were the source population, while all adult >18 years old living in the selected household were the study population. The calculated sample size was 1028. Participants were selected using a multistage sampling technique. A structured, interview-based questionnaire was used to collect the data. The data were entered into Epidata Manager and exported to SPSS for analysis.

### Result

The response rate for this study was 95.4%. The prevalence of poor perception and unfavorable help-seeking intention of mental illness were 45.8%, 95% CI (42.6, 48.9), and 49.5%, 95% CI (46.4, 52.7) respectively. Being rural [AOR = 1.94 (95% CI:(1.41, 2.66)]c, lack of information [AOR = 4.82(95% CI: (3.39,6.83)], exposure to mental illness [AOR = 4.11(95% CI:(2.64,6.38)] were significantly associated with poor perception of mental illness. Also, gating mental illness information [AOR = 0.40 (95% CI: (0.19, 0.83)], and being exposed to mental illness [AOR = 0.56 (95% CI: (0.41, 0.79)] were significantly associated with unfavorable help-seeking intentions for mental illness.

**Funding:** Mizan-Tepi University provided funding for this study. The funders had no role in study design, data collection and analysis, decision to publish, or preparation of the manuscript.

**Competing interests:** he authors have declared that no competing interests exist.

## Conclusion

The high prevalence of poor perceptions and unfavorable help-seeking intentions for mental illness can be minimized through providing tailored information regarding the cause, type, and severity of the problem, particularly in the rural areas.

## Background

Mental illness (MI) is a medical condition that alters a person's thinking, feelings, or behavior, causing distress and difficulty in functioning [1–3]. MI is caused by genetic, biological, environmental, and psychological factors [4–6]. The common types of MI include depression, schizophrenia, attention deficit hyperactivity disorder, autism, and obsessive-compulsive disorder [3, 7, 8].

In 2019, nearly a billion people were living with MI globally [7–10]. The majority of people with MI live in low- and middle-income countries (LMIC) [11, 12]. In Ethiopia, the prevalence of MI is more than 20% [13, 14].

Mental illnesses can be treated with medications, psychotherapy, lifestyle changes, and social support, but poor perceptions of MI contribute to the lack of quality care and stigma of MI sufferers help-seeking [9, 15]. The magnitude of poor perceptions about MI was lower in more affluent nations than LMIC. In Ethiopia, studies show that 40–60% of the community lacks a good perception of mental health problem [16–20].

Socio-demographic and economic factors can influence the perception of MI. High levels of educational status by the peoples can raise awareness about MI and improve the community perception regarding mental health disorders [18, 21, 22]. Also, those who are a rural resident and those with lower socio economic status are more prone to poor perception of MI due to lack of access to health information and health service [19, 23].

In addition to this, religion, culture, beliefs, and traditions can affect individuals to view the cause and management of MI were spiritual or divine power [24, 25]. However, higher levels of community mental health literacy and information about MI improve communities perception towards mental health disorders [18, 20, 22]. Moreover, personal experiences with MI, either through oneself, family, or friends, can lead to a more informed attitude [22, 26].

The high prevalence of poor perception about MI, accompanied by stigma and lack of social support, exposes people to traditional and religious healer [25, 27]. The prevalence of unfavorable help-seeking intention for MI varies across geography; it is 20% in China, but as high as in 75% in Japan, and also in Ethiopia majority of people tend to seek help from traditional healer [19, 20, 28, 29].

However, the majority of studies conducted regarding perceptions of MI and help-seeking intentions for MI in Ethiopia were primarily confined to urban areas where communities have better health care access [16–18, 20]. As a result, the county in general and the SWEPRS in particular lack sufficient and updated information regarding community perception and help-seeking intention for MI. Therefore, this study assessed community perceptions towards MI and help-seeking intentions in SWEPRS.

## Methods and material

### Ethics approval and consent to participate

This study was carried out in accordance with the Helsinki Declaration and with the approval of Mizan Tepi University's Research Ethics Committee with ethics approval number MTU/

00120/2021. The health department of each zone provided a letter of cooperation to get access in the zone. Before beginning the study, all participants provided written informed consent after discussing the aims, purpose, risks and benefits of the study. Furthermore, throughout the study, confidentiality, anonymity, and the freedom to withdraw from the study at any time were respected.

## Study setting and period

This study was done in SWEPRS from March 1ˢᵗ to June 30ᵗʰ,2021. The region is administratively divided in to six zones: namely, Kaffa Zone, Bench-Sheko Zone, Sheka Zone, Westomo Zone, Dawro Zone and Konta Zone. The total population of the region is around 3.5 million. In this area, there are 134 health centers and 836 health posts, two zonal hospitals,10 primary-level hospitals one teaching and referral hospital.

**Study design.**   Community based Analytical cross sectional study design was done.

**Source and study population.**   The source population of this study were all adult population >18 years old residing in southwest Ethiopia. While the study population were all adult population >18 years old residing in the selected kebele for the last six months. Kebele is the smallest legally recognized administrative structure in Ethiopia.

## Inclusion and exclusion criteria

**Inclusion criteria:** All adult population aged >18 years residing in Bench-Sheko, Kaffa, West Omo and Sheka Zone for the last six months were included to the study.

**Exclusion criteria:** Very sick and severally mentally ill patients, those who have communication problem like hearing and speech were excluded from the study.

**Sample size determination.**   The sample size required for this study was calculated for both objectives using a single population proportions and double proportion formula respectively by Open-Epi software.

**Sample size for the first objective.**   The following assumption were taken in to account: proportion of poor community perception towards MI as 50%. $Z_{\alpha/2}$ 95% CL, = 1.96, margin of error is 5%, and design effect 2. The calculated sample size was 768.

$$n = \left(Z_{\frac{\alpha}{2}}\right)^2 \mathrm{P}\left(\frac{1-P}{d^2}\right)$$

Where: n is sample size, Z is standard normal distribution corresponding to significance level at $\alpha = 0.05$, d is margin of error P is anticipated proportion of poor community perception towards MI.

**Sample size for the second objective.**   The sample size calculation for the second objective calculated considering the following assumptions in to account: $Z_{\alpha/2}$ 95% CL, = 1.96, power of 80%, female sex as factor with odds ratio1.75, percent of outcome among exposed was 57% [17] and design effect 2. The calculated sample size was 936 which was larger than sample size calculated for the first objective. Finally adding 10% non- response rate make the final sample size 1028.

**Sampling procedure.**   A multistage sampling technique was used. There were six zones in the SWEPR administratively divided in to district and city administrations. First, four zone were selected randomly using lottery method. From the selected zone a strata of city administration and district administration were formed, then considering the number of these structure 11 districts and four cities were selected. From these selected districts and city administration at least 30% of kebeles were selected randomly. Finally, the sample size was proportionally allocated based on the contribution of the kebele population, and participants

were selected randomly from each kebele using the list of households as a sampling frame (Fig 1).

**Data collection procedures and quality control.**    The data were collected using a structured interview-based questionnaire which was developed after rigorous literature review [17–19]. The questionnaire has six parts: **part I**- about socio-demographic characteristics of the participant, **Part II**- Exposure to MI and history of MI information, **part III**- knowledge about mental illness, **part IV**- Perception about mental illness, **part VI**- perception about cause of MI and **part VII**- help-seeking intention for mental illness.

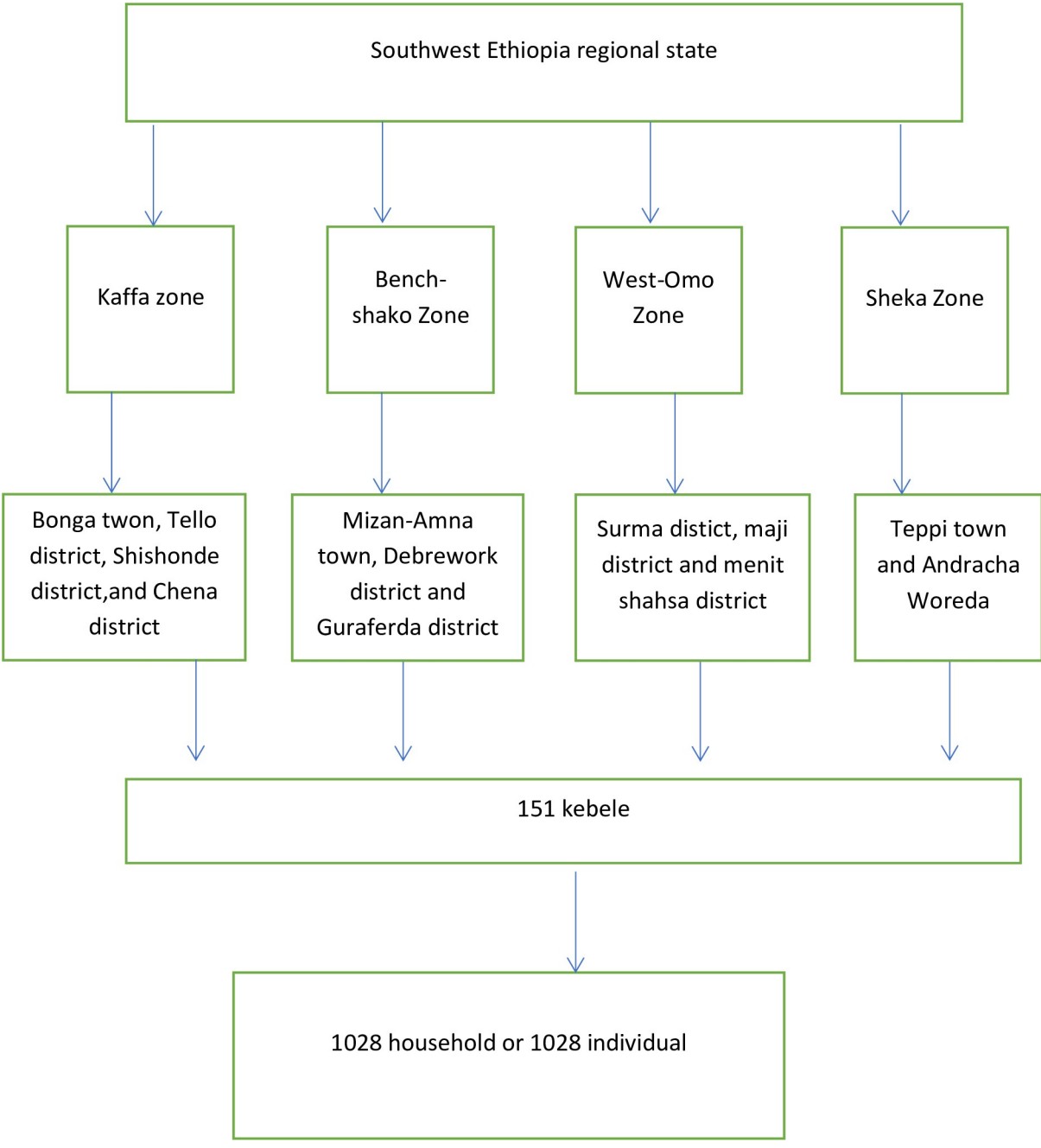

**Fig 1. Sampling distribution.**

## Variables of the study

**Dependent variable:** Community perception towards MI and Help-seeking Intention for MI

**Independent variable:** Socio-demographic factors:—Age, education, residence, occupation, marital status, family size, head of household and income

**Other coverable:** Exposure to MI and history of MI, information, knowledge regarding MI.

## Measurements and operational definition

**Community perception towards MI**: assessed by Community Attitude towards Mentally Ill Questionnaire which was used in previous studies [17–19]. It was measured by using 9 items of five semantic deferential scales: then the sum score was calculated and mean of the sum score was commuted. A cutoff of point equal to and below the mean score considered as having poor perception of MI [19].

**Help-seeking intention for MI:** assessed using General Help-seeking Behavior questionnaire [30]. It has a seven-point Likert scale ranging from extremely unlikely to extremely likely responses. The mean score was calculated for help-seeking behavior questions and a cutoff of point equal to and below the mean score considered as having unfavorable help-seeking intention [19].

**knowledge regarding MI:** assessed using the Mental Health Knowledge Schedule (MAKS) [31]. It has 12 items five-point Likert scale question. Question 6,8, and 12 reverse coded. for knowledge score the higher the score indicates lower knowledge, thus score below and equal to the mean categorized as good knowledge [32].

## Data processing and analysis

Data was cleaned, edited, coded, entered in to Epi data Manager and exported to SPSS software version 25. In SPSS, descriptive statistics were performed to characterize the dependent and independent variables. In analytic statistics, bivariate analysis was done, and candidate variables were selected for multiple variable logistic regression analysis at p value less than 0.25 [33, 34]. Multiple variable logistic regression analysis was done to identify factors associated with poor perception and unfavorable help-seeking intention of MI. An association with a P value less than 0.05 considered as statistically significant association. Finally, an AOR with a 95% CI was reported.

## Result

### Socio demographic characteristics

The response rate for this study was 95.4%. The mean age of the participant was 31.1 ± 9.27. Also, 57.2% of the participants were male. Regarding their education level, 50.2% of the participants cannot read and write. The majority of the participant (56.6%) were Orthodox by religion. Also, 68.6% of the participants were rural resident. The average family monthly income was 1940.45 with SD ± 1564.43 Ethiopian Birr (Table 1).

### Personal experience and information about MI

In this study 98% of the participant have seen people attacked by person with MI, 59% have experienced attack by person with MI, 42% participant participated in caring person with MI, 14% of participant reported they have MI. Also 46% of the participant heard about MI majority of them heard from radio (Table 2).

**Table 1. Socio-demographic characteristics of study participants in southwest Ethiopia, 2021.** (N = 1028).

| Variable | Categories | N (%) |
|---|---|---|
| Age | 18–28 | 461 (47) |
| | 29–38 | 324 (33) |
| | >39 | 196 (20) |
| Gender: | Male | 561(57.2) |
| | Female | 420(42.8) |
| Marital status: | Single | 419(42.7) |
| | Married | 499(50.9) |
| | Divorced | 48(4.9) |
| | Widowed | 15(1.5) |
| Religion: | Protestant | 224(22.8) |
| | Orthodox | 555(56.6) |
| | Muslim | 202(20.6) |
| Educational status | Tertiary | 76(7.7) |
| | Secondary | 196(20.0) |
| | Primary | 218(22.2) |
| | Illiterate | 491(50.1) |
| Occupational status | House wife | 146(14.9) |
| | Government employee | 475(48.4) |
| | Merchant | 176(17.9) |
| | Daily laborer | 145(14.8) |
| | Farmer | 39(4.0) |
| Residence | Urban | 673(68.6) |
| | Rural | 308(31.4) |

## Knowledge of MI

Regarding the overall knowledge of participants about MI, 63.16% of them were knowledgeable (Fig 2). But only 4% of the participants strongly agreed that people with mental health problems want to have paid employment. While 12% of the participants strongly agreed they

**Table 2. Personal experience and information about MI.**

| | | N | % |
|---|---|---|---|
| Have you ever seen a person being scared by a people with mental illness? | Yes | 761 | (78) |
| | No | 21 | (2) |
| Have you ever been attacked by a person with mental illness? | Yes | 578 | (59) |
| | No | 403 | (41) |
| Have ever been participated in caring for mentally ill person | Yes | 415 | (42) |
| | No | 566 | (58) |
| Have you ever had any form of mental illness | Yes | 137 | (14) |
| | No | 844 | (86) |
| Have you ever heard about mental illness from any source within the last one yea | Yes | 454 | (46) |
| | No | 527 | (54) |
| From where did you get the information | None | 517 | (53) |
| | Radio | 336 | (34) |
| | Printed | 37 | (4) |
| | Health institution | 61 | (6) |
| | People | 30 | (3) |

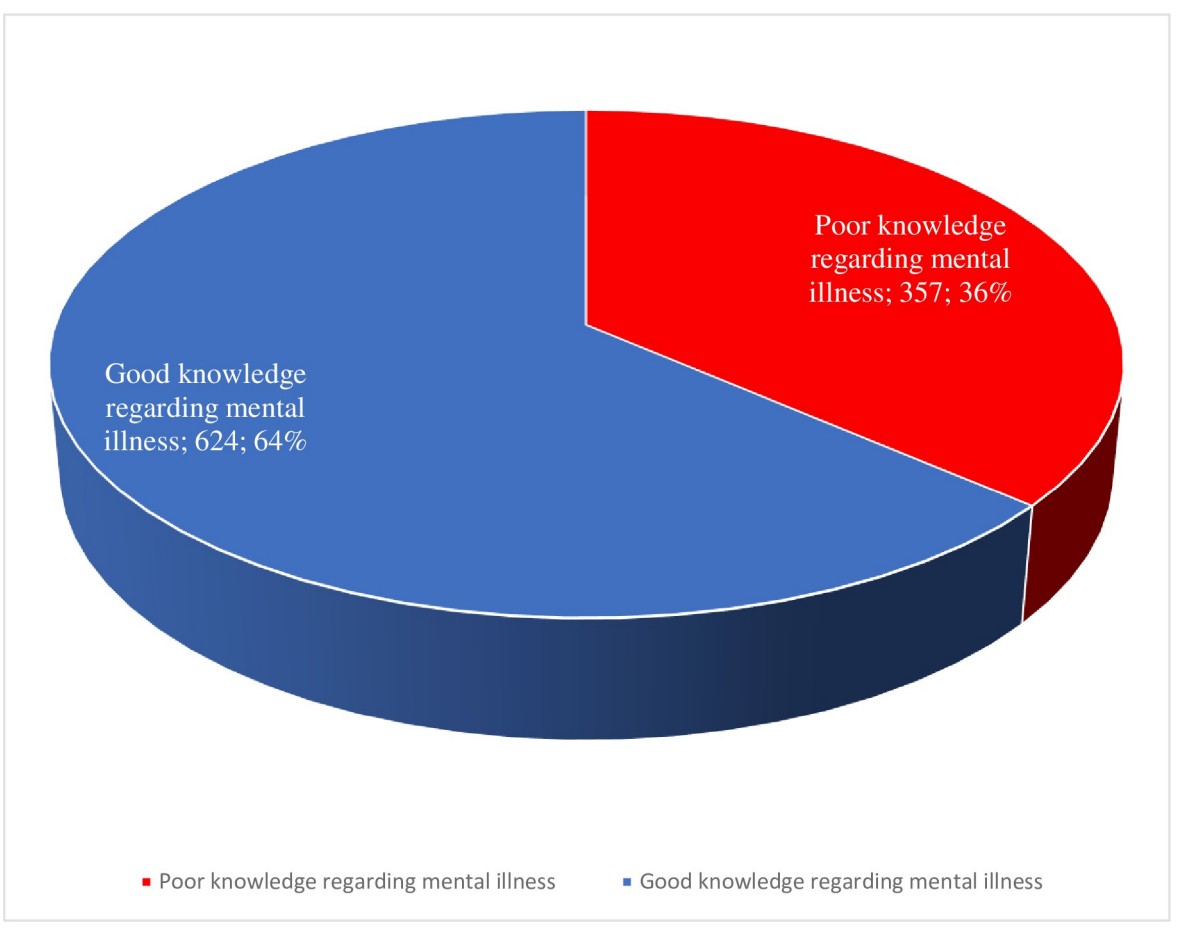

**Fig 2. Knowledge about mental illness among study participants in southwest Ethiopia.**

know what to advise when a friend has MI, 15% strongly agreed medication is an effective treatment for people with MI, and 24% strongly agreed psychotherapy can be an effective treatment for people with mental health problems (Table 3).

**Table 3. Response for psychometric properties of the mental health knowledge schedule question.**

|  | Strongly Disagree | Disagree | Neutral | Agree | Strongly Agree |
|---|---|---|---|---|---|
| Most people with mental health problem want to have paid employment. | 222(23) | 308(31) | 199(20) | 214(22) | 38(4) |
| If a friend had a mental health problem, I know what advice to give them to get | 137(14) | 195(20) | 182(19) | 346(35) | 121(12) |
| Medication can be an effective treatment for people with mental health problem | 87(9) | 137(14) | 260(27) | 347(35) | 150(15) |
| Psychotherapy can be an effective treatment for people with mental health problem | 100(10) | 79(8) | 193(20) | 369(38) | 240(24) |
| People with severe mental health problem can fully recover. | 96(10) | 124(13) | 303(31) | 327(33) | 131(13) |
| Most people with mental health problems go to a healthcare professional to get | 162(17) | 226(23) | 230(23) | 250(25) | 113(12) |
| Depression is a type of mental illness | 120(12) | 152(15) | 227(23) | 367(37) | 115(12) |
| Stress is a type of mental illness | 94(10) | 97(10) | 212(22) | 402(41) | 176(18) |
| Schizophrenia is a type of mental illness | 82(8) | 93(9) | 168(17) | 459(47) | 179(18) |
| Bipolar disorder is a type of mental illness | 80(8) | 57(6) | 144(15) | 456(46) | 244(25) |
| Drug addiction is a type of mental illness | 80(8) | 115(12) | 183(19) | 375(38) | 228(23) |
| Grief is a type of mental illness | 83(8) | 101(10) | 171(17) | 343(35) | 283(29) |

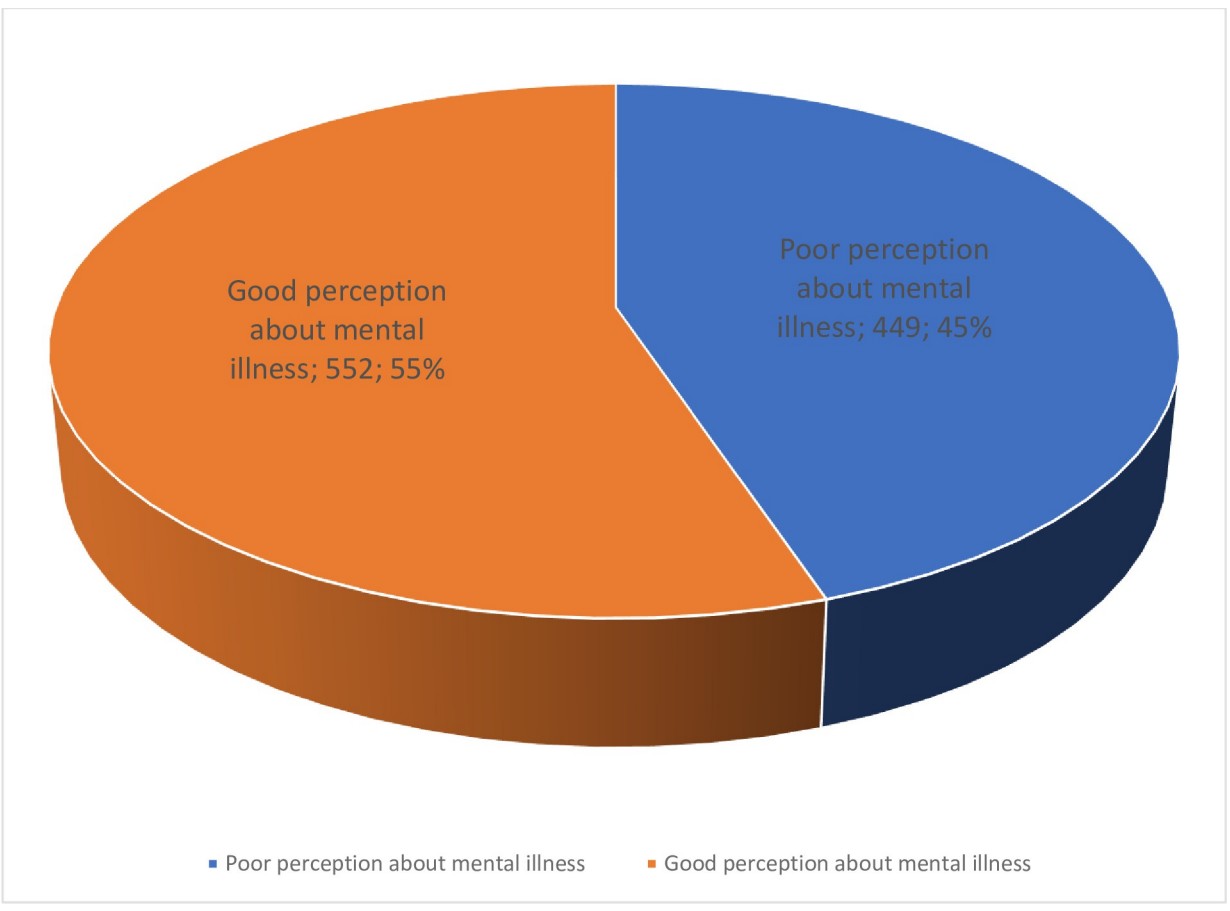

**Fig 3. Community perception about mental illness among study participants in southwest Ethiopia, June, 2021.**

### Community perception about MI

In this study, 45%, with 95% CI (42.6, 48.9) of the participants had poor perceptions about MI (Fig 3). Also, study participants perceived the cause of MI as stress (90.9%), poverty (35.2%), personal weakness (25.9%), God's punishment (389.7%), evil spirits (61.2%), genetic inheritance (306.2%), substance abuse (77.2%), physical illness (51.4%), and germs (26.9%) (Fig 4).

### Factor associated with perception towards MI

In the bivariable logistic regression model, eight variables (age group, gender, educational status, occupational status, residence, information, income, and ill member in the household) were associated with community MI perception at a p value less than 0.25. In multiple-variable regression, three of them showed a statistical association with perception towards MI at a p value less than 0.05.

Accordingly, the odds of poor community perception about MI were 1.94 times higher among rural residents as compared to urban residents [adjusted OR = 1.94 (95% CI: 1.41–2.66)]. Also, the odds of poor community perception about MI were 4.82 times higher among participants who lacked information about MI as compared to their counterparts [adjusted OR = 4.82 (95% CI: 3.39, 6.83)]. Furthermore, the odds of poor community perception about MI were 4.11 times higher among participants who have experienced mentally ill HH members [adjusted OR = 4.11 (95% CI: 2.64–6.38)] (Table 4).

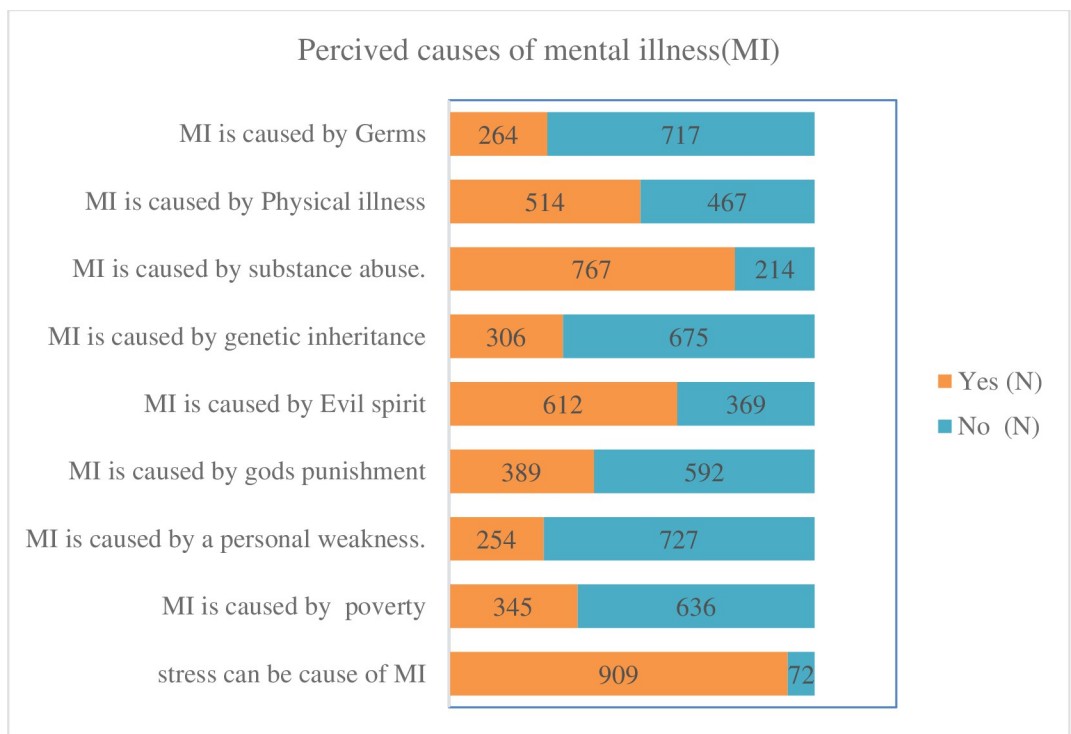

**Fig 4. Perceived causes of mental illness among study participants in southwest Ethiopia, June, 2021.**

### Help-seeking intention

The prevalence of unfavorable help-seeking intention for MI was 49.5%, with 95% CI (46.4, 52.7) (Fig 5).

### Factor associated with help-seeking intention for MI

In the bivariate logistic regression model, three variables namely (residence, income and ill member in the household, have showed significant association at p value less than 0.25. In the multiple variable logistic regression model, gating information about MI from radio and health institution, being exposed to MI in the household, and perception of MI showed statistically significant association at p value less than 0.05 (Table 3).

In this study gating information about MI from radio and health institution decreased the odds of unfavorable help-seeking intention for MI by 60% and 68% respectively [adjusted OR = 0.40 (95% CI: (0.19, 0.83)]. Also, being exposed in MI increased the odds of unfavorable help-seeking intention by 44% [Adjusted OR = 0.56 (95% CI: (0.41, 0.79)]. Moreover, perception of MI was also significantly associated with help-seeking intention for MI [Adjusted OR = 1.36 (95% CI: (1.02, 1.74)] (Table 5).

### Discussion

This study assessed the community's perception of MI and help-seeking intention. The result showed that, 45.8% of the participants have a poor perception of MI. This finding is in consistent with similar study finding in Ethiopia, Dese and Jima Town [17, 19]. However, the finding was higher as compared with similar research done in Ethiopia Gimbi Town [18]. The reason for the difference could be due to socioeconomic and cultural differences among study

**Table 4. Multiple variable logistic regression analysis of community perception towards mental illness in southwest Ethiopia, 2021(N = 1028).**

| | Poor Perception | Good Perception | COR(95% CI) | AOR (95% CI) |
|---|---|---|---|---|
| Age group | | | | |
| 18–28 | 191(41.4) | 270(58.6) | 1 | 1 |
| 29–38 | 154(47.5) | 170(52.5) | 1.531.09,2.16)* | 0.89 (0.63,1.25) |
| >39 | 99(52.1) | 91(47.9) | 1.20(0.84,1.71) | 0.59(0.39,0.89) |
| Gender: | | | | |
| Male | 245(44.0) | 314(56.0) | 1 | |
| Female | 204(48.1) | 218(51.9) | 0.83(.64,1.07)* | 0.82(0.63,1.17) |
| Educational status | | | | |
| Tertiary | 35(46.1) | 41 (53.9) | 1 | 1 |
| Secondary | 105(53.6) | 91(46.4) | 0.74(0.53,1.04) | 0.575(0.31,1.08) |
| Primary | 82(37.6) | 136(62.4) | 1.42(1.03,1.98)* | 0.80 (0.42,1.52) |
| Illiterate | 227(46.2) | 264(53.8) | 0.99(0.61,1.61)* | 0.57 (0.30,1.09) |
| Occupational status | | | | |
| House wife | 226(47.6) | 249(52.4) | 1 | 1 |
| Farmer | 65(44.5) | 81(55.5) | 0.88(.61,1.28) | 0.84(0.50,1.40) |
| Daily laborer | 65(36.9) | 111(63.1) | 1.37(0.87,2.11)* | 1.23 (0.70,2.16) |
| Merchant | 82(56.6) | 63(43.4) | 0.62(0.38,0.98)* | .629(0.35,1.11) |
| Government | 11(28.2) | 28(71.8) | 2.04(0.94,4.44)* | 1.81 (0.75,4.37) |
| Residence | | | | |
| Urban | 192(62.3) | 116(37.7) | 0.37(0.28,0.49)* | 0.35(0.31,0.66)** |
| Rural | 257(38.2) | 416(61.8) | 1 | 1 |
| Information | | | | |
| No | 150(33) | 304(57) | .37(.29, .488)* | 4.82(3.39,6.83)** |
| Yes | 299(67) | 228(43) | 1 | 1 |
| Presence of mentally ill person in the household | | | | |
| Yes | 34(19.0) | 145(81.0) | 4.57(3.07,6.81)* | 4.11(2.64,6.38)** |
| No | 415(51.7) | 387(48.3) | 1 | 1 |
| Monthly Income | | | | |
| <1000 | 226(50) | 308(58) | 1 | 1 |
| 1001–2999 | 55(12) | 69(13) | 1.41(.99, 2.04)* | 3.766(2.28,6.21) |
| 3000–4999 | 91(20) | 81(15) | 1.30 (.81, 2.10) | 2.35(1.52,3.63) |
| >5000 | 77(17) | 74(14) | .92 (.59, 1.44) | 5.60 (3.36,9.32) |
| Knowledge about mental illness | | | | |
| Poor knowledge | 310(69) | 394(74) | 1.28(.97, 1.69) | |
| Good knowledge | 139(31) | 138(26) | 1 | |

*≤0.25

**≤0.05

participants; because previous studies done in Ethiopia assessed the perception of MI in the urban population only [18].

In addition to this this study found factors associated with the high prevalence of poor perception of MI. Accordingly, the odds of poor community perception about MI were found to be decreased by 65% among urban residents as compared to rural residents. The finding was inconsistent with study result in Ethiopia [18]. The urban residents could have relatively better education and access to health information, this helps them to have improved perception about health problems.

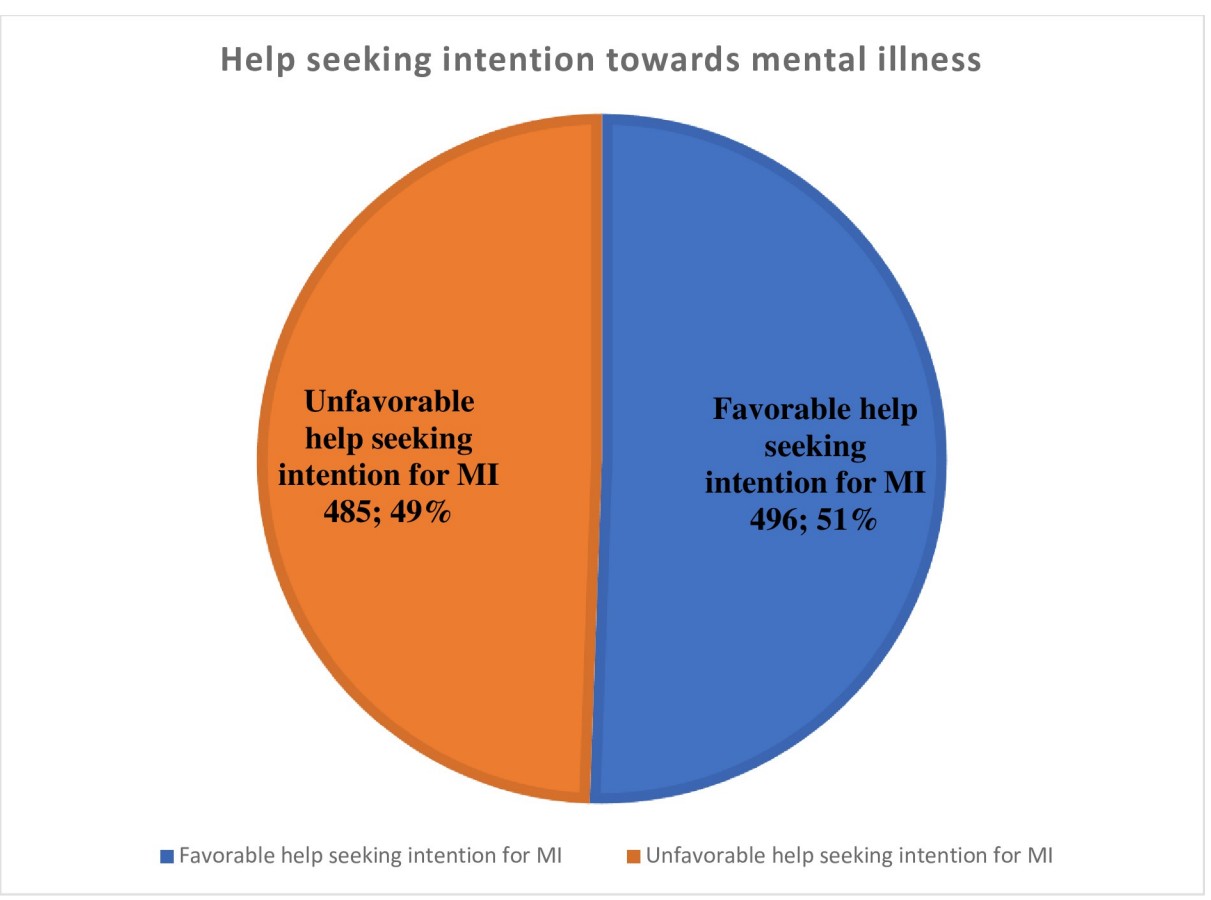

**Fig 5. Help-seeking intention towards mental illness among study participants in Southwest Ethiopia, June, 2021.**

Also, the odds of poor perception about MI were 4.82 times higher among participants who lacked information about MI. The finding is in line with previous studies [18, 20]. This shows providing timely and appropriate information can improve how peoples perceive the health problem around them.

Moreover, this study found that the odds of poor community perception of MI were 4.11 times higher among participants who have mentally ill family member. This might happen due to different burdens, including financial and psychological pressure. Again, families who have mentally ill persons could get wrong information while searching for support from their locality or usually traditional healers.

Regarding the perceived cause of mental illness, a significant proportion of the participants pointed out that supernatural power and stress were the most common reasons for MI, followed by substance use, personal weakness, God's punishment, the evil spirit, genetic inheritance, and substance abuse. Physical illness were the least reported reason [35, 36]. However, the finding was contrary to a study report from the western world that indicated psychosocial factors,, environmental stressors and traumatic experiences as major cause of MI [37].

In this study, 49.5% of the participants lacked favorable help-seeking intentions for MI. This finding is in line with study finding in Ethiopia and in Saudi Arabia [25]. However, the finding was lower when compared with the study finding from Jimma Zone which shows the prevalence of unfavorable help-seeking intention was 60% [19].

**Table 5. Multiple variable logistic regression analysis of help-seeking intention for mental illness in southwest Ethiopia, 2021(N = 1028).**

| | Favorable help-seeking intention | unfavorable help-seeking intention | COR (95% CI) | AOR (95% CI) |
|---|---|---|---|---|
| Age group | | | | |
| 18–28 | 246 | 215 | 1 | |
| 29–38 | 153 | 171 | 1.28(.96,1.70) | |
| >39 | 83 | 107 | 1.48(1.05,2.07) | |
| Gender | | | | |
| Male | 285 | 274 | 1 | |
| Female | 201 | 221 | 1.14(.88,1.47) | |
| Educational status | | | | |
| Tertiary | 36 | 40 | 1 | |
| Primary | 99 | 97 | .88(.52,1.45) | |
| Secondary | 99 | 119 | 1.08(.64,1.82 | |
| Can't read | 252 | 239 | .85(.52,1.38) | |
| Occupational status | | | | |
| Employee | 70 | 76 | 1 | |
| Housewife | 236 | 239 | .93(.64,1.35) | |
| Merchant | 86 | 90 | .96(.62,1.49) | |
| Daily laborer | 70 | 75 | .99(.62,1.56) | |
| Farmer | 24 | 15 | .57(.28,1.18) | |
| Residence | | | | |
| Rural | 349 | 324 | 1 | 1 |
| Urban | 137 | 171 | 1.34(1.03,1.76)* | 1.45(1.10,1.92) |
| Exposure to mental illness | | | | |
| Yes | 71 | 108 | 1 | 1 |
| No | 415 | 387 | .61(.44,.85)* | .56(.41,.79)** |
| Information | | | | |
| Yes | 217 | 237 | 1 | |
| No | 269 | 258 | .87(.68,1.12) | |
| Source of information | | | | |
| None | 246 | 271 | 1 | 1 |
| radio | 162 | 174 | .38(.18,79)* | .40(0.19,0.83)** |
| printed | 26 | 11 | .98(.74,1.28)* | .95(0.71,1.25) |
| health institution | 29 | 32 | .27(.11,.65)* | .28(0.12,0.67)** |
| people | 23 | 7 | 1.00(.58,1.70)* | 1.02(0.60,1.75) |
| Monthly income | | | | |
| <1000 | 255 | 279 | | |
| 1001–2999 | 66 | 58 | .80(.54,1.18) | |
| 3000–4999 | 95 | 77 | .74(.52,1.04) | |
| >5000 | 70 | 81 | 1.05(.73,1.51 | |
| Social support | | | | |
| Poor | 196 | 183 | 1 | |
| Moderate | 217 | 237 | 1.17(.89,1.53) | |
| Strong | 73 | 75 | 1.10(.75,1.60) | |
| Perception | | | | |
| poor perception | 233 | 216 | 1.20(.93,155)* | 1.36(1.02,1.74)** |
| good perception | 251 | 281 | 1 | 1 |

*<0.25

**<0.05

Also, the odds of unfavorable help-seeking intention decreased by 60% and 68%, respectively, among study participants who received information about MI from the radio and a health facility. This is because information can influence an individual's decision about where to get treatment [20, 32].

In addition, being exposed to MI increased the odds of unfavorable help-seeking intentions by 44%. Moreover, the odds of unfavorable help-seeking were 1.36 times high among participant who lacked good perception of MI [32]. Help-seeking from traditional and religious healer is primarily related to the assumed cause of MI being supernatural power or an evil spirit, but it should not be overlooked that a lack of health facilities that provide mental health services in the area is also one of the reasons why people prefer conventional treatment.

## Limitation of the study

One of the limitations of this study was social desirability bias, since the data collection method was face-to-face interviews with individuals who may have responded socially acceptable answer. Secondly, this study did not assess perception of the community towards specific types of MI. It is possible that some subtle difference exists on perception of severe mental illness. To minimize the potential bias the data collection tools were prepared cautiously and also data collector were trained to clarify the aim of the study to the participant. But the strong side of this study is that, it tried to assess community perceptions towards MI and help-seeking intention for MI from multiple centers.

## Conclusion

This study showed, nearly half of the community in the study area have poor perception and unfavorable help-seeking intention for MI. Being from rural resident, having information about MI, and experiencing MI were significantly associated factors with poor perception of MI. Furthermore, getting information from media, and health facilities, experiencing MI, and poor perception of MI were significantly associated factors with unfavorable help-seeking intention for MI.

Therefore, providing appropriate and tailored information regarding the cause, type and severity of MI focusing on the rural residents and who have experienced MI were important to improve the community MI perceptions and help-seeking intentions in the study area. This can be achieved through integrating the community mental health education initiatives and community mental health services in to existing Ethiopia's primary health care system.

## Supporting information

**S1 Dataset.**
(XLSX)

**S1 Questionnaire.**
(DOCX)

## Acknowledgments

We are grateful for zonal health departments of Bench-Shako, West-Omo, Sheka, and Kaffa, as well as all Health district offices, for their cooperation. Secondly, we would like to extend our heartfelt appreciation to the study participants for their willingness to be interviewed.

## Author Contributions

**Conceptualization:** Zenebu Muche.

**Data curation:** Dawit Getachew, Gebremeskel Mesafint.

**Formal analysis:** Dawit Getachew, Nahom Solomon.

**Funding acquisition:** Dawit Getachew.

**Investigation:** Dawit Getachew, Gebremeskel Mesafint, Nahom Solomon, Zenebu Muche, Sewagegn Demelash.

**Methodology:** Dawit Getachew.

**Project administration:** Dawit Getachew.

**Software:** Dawit Getachew.

**Supervision:** Dawit Getachew, Gebremeskel Mesafint, Zenebu Muche.

**Validation:** Dawit Getachew, Gebremeskel Mesafint.

**Visualization:** Dawit Getachew.

**Writing – original draft:** Dawit Getachew, Gebremeskel Mesafint, Nahom Solomon, Kidus Yenealem, Zenebu Muche, Sewagegn Demelash.

**Writing – review & editing:** Dawit Getachew, Gebremeskel Mesafint, Nahom Solomon, Kidus Yenealem, Zenebu Muche, Sewagegn Demelash.

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
