## [Decision Letter · Decision Letter 0]

9 Jun 2024

PONE-D-23-13743Community perception and help-seeking behavior towards mental illness in Southwest EthiopiaPLOS ONE

Dear Dr. Getachew ,

Thank you for submitting your manuscript to PLOS ONE. After careful consideration, we feel that it has merit but does not fully meet PLOS ONE’s publication criteria as it currently stands. Therefore, we invite you to submit a revised version of the manuscript that addresses the points raised during the review process. 

We look forward to receiving your revised manuscript.

Kind regards,

Firomsa Bekele Negera, Msc

Academic Editor

PLOS ONE

2. PLOS ONE does not copy edit accepted manuscripts (https://journals.plos.org/plosone/s/criteria-for-publication#loc-5). To that effect, please ensure that your submission is free of typos and grammatical errors.

3. In the online submission form, you indicated that [IThe data set can be accessible from the corresponding author up on reasonable request.]. 

Additional Editor Comments (if provided):

Reviewers' comments:

Reviewer's Responses to Questions

**Comments to the Author**

1. Is the manuscript technically sound, and do the data support the conclusions?

Reviewer #1: Yes

Reviewer #2: Yes

Reviewer #3: Partly

2. Has the statistical analysis been performed appropriately and rigorously? 

Reviewer #1: No

Reviewer #2: Yes

Reviewer #3: Yes

3. Have the authors made all data underlying the findings in their manuscript fully available?

Reviewer #1: Yes

Reviewer #2: Yes

Reviewer #3: No

4. Is the manuscript presented in an intelligible fashion and written in standard English?

Reviewer #1: Yes

Reviewer #2: Yes

Reviewer #3: No

5. Review Comments to the Author

Reviewer #1: In general the title of research is talk about perception community and help seeking of MI, but the researcher is talk about factors relate to MI, there is mismatch between title and discussion . It need some modification

Reviewer #2: Decision on Community perception and help-seeking behavior towards mental illness in Southwest Ethiopia

Dawit Getachew, MPH

Comments from Reviewer

Dear Authors,

Thank you for submitting the manuscript to the PLOS ONE journal. The manuscript is well written, though modifications and revisions are needed. The professional language edition is warranted as the manuscript needs synthesis and revision.

General: There are great inconsistencies and grammar issues.

There is a great discrepancy between the study topic and the results reported. The results are about recurrences of mental illness and its associated factors. Nothing was reported about the poor health seeking behavior of MI patients. Authors are suggested to update the study title to make it representative of the study results.

Title: The title is not specific, the author tried to address the health seeking behavior, and associated factors of MI at the same time which are unrelated and could be independent study topics. I suggest modifying the title to focus on either Health seeking behavior with associated factors. The title of the study also goes beyond its expected scope. The study was conducted in a single center (Mizan Tepi). However, the author(s) generalized the results for all Southwest Ethiopia. I suggest modifying the study scope. The title seems to that the study was conducted in single centers in Southwest Ethiopia; however, it is not associated with the study area. Hence, change the study area to a specific area.

Abstract

Avoid abbreviations in the abstract.

This study was aimed to assess community perception and help seeking Intention towards mental illness in Southwest Ethiopia.

Background: The abstract's background should discuss the background of the problem in the country context and show the study gaps. I recommend the author(s) to update it.

Make I lower case and rewrite as intention.

Avoid Bold. Instead Make A community-based

Results: The results of the abstract were not written in line with the study objectives. Most of the results have a very wide confidence interval. How can the authors justify the wider CL?

Arrange the keywords in alphabetical order i.e Keywords: Mental illness, Perception, help-seeking intention, Southwest Ethiopia

Introduction

Indicate the prevalence of Mental illness in Ethiopia by number, not by prevalence. . fore example Depression, schizophrenia, attention deficit hyperactivity disorder (ADHD).Remove full-stop and make sentence case i.e. For example,

As for the knowledge gap, you have mentioned the delivery and implementation of the mental health

service were not successful in this study area. Would you indicate the number of studies published and the summary of their findings? Plus merge your citations. Therefore majority of the community prefers to seek treatment either from religious or spiritual healers for MI [15-21] [22]. illness[32, 34, 35] [24, 26]. merge your citations.

The last sentence of the introduction should state the aim of the study. I suggest Adding the research question.

Methods

Methods: the methods part of the abstract did not indicate the sample size, and sampling technique used, and how the results of regression analysis were interpreted (the CI, OR, P-value). The author(s) mentioned, “data was collected by data collectors”. Who were those data collectors? It needs to be specified.

Study area: Make sure the information is valid and correct. The number of people with mental illness should be mentioned.

Separate the study design and population. The inclusion and exclusion criteria for the participants should be indicated.

Sample size determination

It is not possible to assume that mental illness is directly related to medication. So why did you assume the p-of poor community perception towards MI as? There is study conducted in Ghimbi town,Ethiopia

Before you apply FPCF, the estimated sample size should be indicated. Why didn’t you include the estimate as it bears benefit for inference?

Why didn’t you use a simple random or systematic sampling technique?

Please indicate the psychometric properties of the MI.

Why post-data collection translation of the scale was not performed?

In order to give the participant confidentiality to their response, the interview will be conducted in separated room with protected privacy.Better to put under ethical consideration. But for knowledge score the higher the score indicate lower knowledge, thus score below and equal to the mean categorized as good knowledge. Why?

Operational Definitions: Use appropriate punctuation.

Data analysis

Please cite the literature that supports the p-value for bivariate logistic regression.

Results

What was the response rate?

Rather than saying ‘men’ it is better to say ‘males’

The description of socio-demographic characteristics needs to be longer; would you elaborate more?

Clinical characteristics of the study participants

Give an interpretation of the findings rather than simply mentioning the results. For example. Nearly, more than and etc.

Factor associated with Help seeking intention towards MI

Is there any justification for no associated factors with poor perception and help seeking intention towards mental illness?

For a p-value of 0.00, please report as P< 0.001. All p-values should be reported in three digits.

Discussion

The discussion should start with the aim of the study and then mention the significant findings of the study. A total of 561 people with MI participated in this ambidirectional cross-sectional study. What does it mean?

Paragraph 1; It may be beneficial if you rewrite.

This discrepancy in the outcome can be the result of using a different data collection technique or sample size. Is there any evidence showing this conclusion?

Regarding the factors associated, you should only discuss those significantly associated.

It would be better if you cite to make strong arguments.

What are the implications of the study?

Limitations of the study

You have listed several limitations. Do you think these limitations affect the validity of your study? I think there are strengths of the study; hence, need to add.

Conclusion

Did you calculate the proportion or the score?

Is there any recommendation?

References

Revise references. Follow the Vancouver citation style.

Reviewer #3: PONE-D-23-13743

Community perception and help-seeking behavior towards mental illness in Southwest Ethiopia.

First I would like to say thank you for the opportunity to review the paper. The title is so interesting and I have points that need clarifications and improvements.

General comments

1. Unnecessary capitalizations. E.g.: line 25, and others and do proofreading accordingly.

2. Background: editorial comments like unnecessary full stops, and inappropriate citations.

3. The data collection tool needs citation under the method: data collection tool section.

A. Under the background, your study gap is not enough to conduct the study and needs improvement.

B. Sample size: Adding non-response or multiplying by design effect…which comes first? And again your sample size seems incorrect. I need clarification.

C. Sample size: clearly show the necessary steps for both the first and second objectives.

D. What is your reference regarding variable categorization, for example under socio-demographic characteristics like age?

E. Table 3: why it is necessary to put variables like socio-demographic characteristics in the table?

F. Strength and limitation:

Study design cannot be a limitation of your study.

The title by itself cannot be the strength of the study.

G. The result in the figure and the document is different and it seems fabricated.

6. PLOS authors have the option to publish the peer review history of their article (what does this mean?). If published, this will include your full peer review and any attached files.

Reviewer #1: No

Reviewer #2: **Yes: **Dinka Dugassa Iticha

Reviewer #3: No

---

## [Author Response · Author response to Decision Letter 0]

24 Jul 2024

Point by point response to the reviewer comment 

Editor comment 

PONE-D-23-13743

Community perception and help-seeking behavior towards mental illness in Southwest Ethiopia

PLOS ONE

Dear Dr. Getachew ,

Thank you for submitting your manuscript to PLOS ONE. After careful consideration, we feel that it has merit but does not fully meet PLOS ONE’s publication criteria as it currently stands. Therefore, we invite you to submit a revised version of the manuscript that addresses the points raised during the review process.

Author response: thank you dear editor, we extremely appreciate your support and also all reviewers for their comments and question. In the revised manuscript we have tried to address all the comments and points raised by the editors, reviewers and also the journal requirement. 

Author response: thank you dear editor regarding the financial disclosure we are waivered by the request we made on the first submission and we are not going to change that.

Author response: thank you dear editor we have prepared and submitted the figure file as per PLOS guideline for figure file submission. 

Author response: thankyou dear editors this is not applicable on the current manuscript. 

We look forward to receiving your revised manuscript.

Kind regards,

Firomsa Bekele Negera, Msc

Academic Editor

PLOS ONE

Author response: Thank you dear editor we have prepared the revised manuscript based on the journal requirement using The PLOS ONE style templates. 

2. PLOS ONE does not copy edit accepted manuscripts (https://journals.plos.org/plosone/s/criteria-for-publication#loc-5). To that effect, please ensure that your submission is free of typos and grammatical errors.

Author response: Thank you dear editor we have copy edited the manuscript to make it free of typos and grammatical error. 

Author response: Thank you dear reviewer we have accepted the comment and corrected on the revised manuscript.

Author response: Thank you dear reviewer we have accepted the comment and corrected on the revised manuscript.

3. In the online submission form, you indicated that [IThe data set can be accessible from the corresponding author up on reasonable request.]. 

Author response: Thank you dear editor we have uploaded the data set as supplementary material

Author response: thank you dear editor we have placed the ethics statement in methods and material section.

Reviewer 1 comment. 

PLOS ONE

Community perception and help-seeking behavior towards mental illness in Southwest Ethiopia

Comments 

Author response: Thank you dear reviewer, we have accepted the comment and corrected the title in the revised manuscript. 

‘Community Perception towards Mental Illness and Help Seeking Intention in Southwest Ethiopian Peoples Regional State’ (page number 1, line number 1-2).

1. Title should be written in capital for each word

Author response: Thank you dear reviewer, we have accepted the comment and corrected the title on the revised manuscript. ‘Community Perception towards Mental Illness and Help Seeking Intention in Southwest Ethiopian Peoples Regional State’ (Page number 1, Line number 1-2).

2. Key word indicates about variable, so why southwestern Ethiopia is come to as a key word? 

Author response: Thank you dear reviewer, we have omitted southwest Ethiopia from the keyword and we have included pertinent variable as a keyword. 

Keywords: Help seeking intention, Mental illness information, Perception of Mental illness. (Page number 2, Line number 47). 

3. In abstract there is no clear statement of design, method and recommendation there is some ambiguity. 

Author response: Thank you dear reviewer, we have rewritten the Abstract by clearly describing the design, methods and implication of the finding and recommendations.

(Page number 2, Line number 36-46).

4. In introduction part there is inconstant word fluency and it does not address issue going be to study conducted 

Author response: Thankyou dear reviewer we have accepted your comment and we have copy edited the introduction section of the manuscript. Therefore, the current study can address the issue studied more clearly. (Page number 3, Line number 49-71).

5. There is no clear contextual study from general to specific (no local study – global) 

Author response: Thankyou dear reviewer we have shown the context of the study from general to specific in the revised manuscript. (Page number 3, Line number 49-71). 

6. There is no clear gap is indicated. what the previous study was reviled about these research 

Author response: Thankyou dear reviewer we have included what previous research reviled about communities’ perception towards mental illness and help seeking behaviour for mental illness. 

7. In the Methodology part researcher not show the design approach, qual or quan or mixed and sampling is not clear 

Author response: Thank you dear reviewer we have accepted your comment in the methods and material section of revised manuscript we have clearly putted the design of this research was quantitative. (Page number 4, Line number 80).

Also, we have described the sampling technique was a multistage sampling technique and we have showed how study participant were selected by including a sampling distribution scheme. (Page number 5, Line number 109-112).

8. In data collection why interview is needed for large population? since the study is for large population?

Author response: Thank you dear reviewer we have used interview method because majority of the participant cannot read and write (Page number 7, Line number 112).

Reviewer comment: In general, the title of research is talk about perception community and help seeking of MI, but the researcher is talk about factors relate to MI, there is mismatch between title and discussion. It needs some modification 

Author response: Thank you dear reviewer we have accepted your comment and we have edited the manuscript by focusing on our primary and secondary outcome perception community and help seeking of MI. (Page number 3, Line number 49-71). 

Reviewer 2 comment. 

Decision on Community perception and help-seeking behavior towards mental illness in Southwest Ethiopia

Dawit Getachew, MPH

Comments from Reviewer

Dear Authors,

Thank you for submitting the manuscript to the PLOS ONE journal. The manuscript is well written, though modifications and revisions are needed. The professional language edition is warranted as the manuscript needs synthesis and revision.

Author response: Thank you dear reviewer we accept the comment we have copy edited the manuscript to correct all grammatical and spelling error that obscured the easy communication and correct message of the paragraph. 

Comments from Reviewer: General: There are great inconsistencies and grammar issues.

Author response: Thank you dear reviewer we have corrected all the inconsistencies and grammatical error in the revised manuscript.

Comments from Reviewer: There is a great discrepancy between the study topic and the results reported. The results are about recurrences of mental illness and its associated factors. Nothing was reported about the poor health seeking behavior of MI patients. Authors are suggested to update the study title to make it representative of the study results.

Author response: Thank you dear reviewer, we appreciate the comment and we wan to apologies for the inconvenience. However, this study assessed healthy individuals regarding their perception and their help seeking intention towards mental illness. But we have copy edited the whole manuscript to make clear the information conveyed. 

Comments from Reviewer: Title: The title is not specific, the author tried to address the health seeking behavior, and associated factors of MI at the same time which are unrelated and could be independent study topics. I suggest modifying the title to focus on either Health seeking behavior with associated factors. The title of the study also goes beyond its expected scope. The study was conducted in a single center (Mizan Tepi). However, the author(s) generalized the results for all Southwest Ethiopia. I suggest modifying the study scope. The title seems to that the study was conducted in single centers in Southwest Ethiopia; however, it is not associated with the study area. Hence, change the study area to a specific area.

Author response: Thank you dear reviewer, we appreciate your genuine comment. Regarding the title of this study, it is about Community Perception towards Mental Illness and Help Seeking Intention among the general population. We agree with you the manuscript can be written as independent article. However, it’s also verry advantageous for planner, policy maker and even for researcher to find both information at one place. 

Also, this study was a community based multi-centre which address four zone of Southwest Ethiopian people’s regional state including 11 district administration and four city administrations. The finding can represent all 57 administrative structure of the six zone of Southwest Ethiopian people’s regional state as we have used multistage sampling and also multiplied the sample size with design effect. 

 Comments from Reviewer: 

Abstract

 Avoid abbreviations in the abstract. 

Author response: Thank you dear reviewer we have removed all the abbreviation from the abstract section of the manuscript. 

Comments from Reviewer:

 This study was aimed to assess community perception and help seeking Intention towards mental illness in Southwest Ethiopia. 

Author response: Yes, dear reviewer this study was aimed to assess community perception and help seeking Intention towards mental illness in Southwest Ethiopia. 

Comments from Reviewer:

 Background: The abstract's background should discuss the background of the problem in the country context and show the study gaps. I recommend the author(s) to update it.

Author response: Thank you dear reviewer we have updated the back ground of the abstract section by discussing the problem under investigation. 

Comments from Reviewer:

 Make I lower case and rewrite as intention.

 Avoid Bold. Instead Make A community-based 

Author response: thank you dear reviewer we apologies for all the incorrect spelling we have correct in the revised manuscript. We have also omitted unnecessary Bold case. 

Comments from Reviewer:

 Results: The results of the abstract were not written in line with the study objectives. Most of the results have a very wide confidence interval. How can the authors justify the wider CL?

Author response: Thank you dear reviewer, we have corrected the result subsection of the abstract. We have showed the magnitude of community perception and help seeking intention towards mental illness in the study area. Then we have shown the factors associated with community perception and help seeking intention. (Page number 2, Line number 36-43).

Comments from Reviewer:

 Arrange the keywords in alphabetical order i.e Keywords: Mental illness, Perception, help-seeking intention, Southwest Ethiopia

Author response: thank you dear reviewer for your comment we have arranged the keyword in alphabetical order, in addition we have updated the key word as per reviewer one comment

Keywords: Help seeking intention, Mental illness information, Perception of Mental illness. (Page number 2, Line number 47). 

Comments from Reviewer:

Introduction

Indicate the prevalence of Mental illness in Ethiopia by number, not by prevalence. . fore example Depression, schizophrenia, attention deficit hyperactivity disorder (ADHD).Remove full-stop and make sentence case i.e. For example,

As for the knowledge gap, you have mentioned the delivery and implementation of the mental health service were not successful in this study area. Would you indicate the number of studies published and the summary of their findings? Plus merge your citations. Therefore majority of the community prefers to seek treatment either from religious or spiritual healers for MI [15-21] [22]. illness[32, 34, 35] [24, 26]. merge your citations.

Author response: thank you dear reviewer We have presented the prevalence of Mental illness disorders in Ethiopia by number as per your recommendation.

 Comments from Reviewer:

The last sentence of the introduction should state the aim of the study. I suggest Adding the research question.

Author response: Thankyou dear reviewer we have accepted the comment and we have added the aim of the study on the last line of the introduction section. ‘There for this study aimed to assess community perception towards MI and communities help seeki

---

## [Decision Letter · Decision Letter 1]

8 Aug 2024

PONE-D-23-13743R1Community Perception towards Mental Illness and Help Seeking Intention in Southwest Ethiopian Peoples Regional StatePLOS ONE

Dear Dr. Getachew ,

Thank you for submitting your manuscript to PLOS ONE. After careful consideration, we feel that it has merit but does not fully meet PLOS ONE’s publication criteria as it currently stands. Therefore, we invite you to submit a revised version of the manuscript that addresses the points raised during the review process.

We look forward to receiving your revised manuscript.

Kind regards,

Firomsa Bekele

Academic Editor

PLOS ONE

Journal Requirements:

Additional Editor Comments:

dear authors, the manuscript needs correction of English grammar

Reviewers' comments:

Reviewer's Responses to Questions

**Comments to the Author**

1. If the authors have adequately addressed your comments raised in a previous round of review and you feel that this manuscript is now acceptable for publication, you may indicate that here to bypass the “Comments to the Author” section, enter your conflict of interest statement in the “Confidential to Editor” section, and submit your "Accept" recommendation.

Reviewer #2: All comments have been addressed

2. Is the manuscript technically sound, and do the data support the conclusions?

Reviewer #2: Yes

3. Has the statistical analysis been performed appropriately and rigorously? 

Reviewer #2: Yes

4. Have the authors made all data underlying the findings in their manuscript fully available?

Reviewer #2: Yes

5. Is the manuscript presented in an intelligible fashion and written in standard English?

Reviewer #2: Yes

6. Review Comments to the Author

Reviewer #2: (No Response)

7. PLOS authors have the option to publish the peer review history of their article (what does this mean?). If published, this will include your full peer review and any attached files.

Reviewer #2: **Yes: **Dinka Dugassa Iticha

---

## [Author Response · Author response to Decision Letter 1]

15 Aug 2024

PONE-D-23-13743R1

Community Perception towards Mental Illness and Help Seeking Intention in Southwest Ethiopian Peoples Regional State

PLOS ONE

Dear Dr. Getachew ,

Thank you for submitting your manuscript to PLOS ONE. After careful consideration, we feel that it has merit but does not fully meet PLOS ONE’s publication criteria as it currently stands. Therefore, we invite you to submit a revised version of the manuscript that addresses the points raised during the review process.

Author response: Thank you dear editor, we acknowledge for support and guidance from the reviewers and editors. We have revised the manuscript to meet the journal requirement and the reviewers comment as well as the editor’s suggestion. 

Author response: thank you dear editor regarding the financial disclosure we are waivered by the request we made on the first submission and we are not going to change that.

Author response: thank you dear, but this is not applicable to this manuscript

We look forward to receiving your revised manuscript.

Kind regards,

Firomsa Bekele

Academic Editor

PLOS ONE

Journal Requirements:

Author response: Thank you dear editor, we have tried to identify if the reference list contain a retracted reference using retracted watch database retractiondatabase.org, we couldn’t find any of the listed reference were retracted. However, we have accepted the comment and updated the reference.

Additional Editor Comments:

dear authors, the manuscript needs correction of English grammar

Author response: Thank you dear editor we have accepted the comment, we copy edited the manuscript to reduce the grammatical and spelling error, to make the manuscript suitable for publication. 

Reviewers' comments:

Reviewer's Responses to Questions

Comments to the Author

1. If the authors have adequately addressed your comments raised in a previous round of review and you feel that this manuscript is now acceptable for publication, you may indicate that here to bypass the “Comments to the Author” section, enter your conflict of interest statement in the “Confidential to Editor” section, and submit your "Accept" recommendation.

Reviewer #2: All comments have been addressed

2. Is the manuscript technically sound, and do the data support the conclusions?

Reviewer #2: Yes

3. Has the statistical analysis been performed appropriately and rigorously?

Reviewer #2: Yes

4. Have the authors made all data underlying the findings in their manuscript fully available?

Reviewer #2: Yes

5. Is the manuscript presented in an intelligible fashion and written in standard English?

Reviewer #2: Yes

6. Review Comments to the Author

Reviewer #2: (No Response)

7. PLOS authors have the option to publish the peer review history of their article (what does this mean?). If published, this will include your full peer review and any attached files.

Do you want your identity to be public for this peer review? For information about this choice, including consent withdrawal, please see our Privacy Policy.

Reviewer #2: Yes: Dinka Dugassa Iticha

Author response: Thank you dear editor we have converted the figure file using Preflight Analysis and Conversion Engine (PACE).

---

## [Editor Report · Acceptance letter]

4 Oct 2024

PONE-D-23-13743R2 

PLOS ONE

Dear Dr. Getachew , 

I'm pleased to inform you that your manuscript has been deemed suitable for publication in PLOS ONE. Congratulations! Your manuscript is now being handed over to our production team.

Kind regards, 

on behalf of

Dr. Firomsa Bekele 

Academic Editor

PLOS ONE